# Thermal control of sequential on-surface transformation of a hydrocarbon molecule on a copper surface

Shigeki Kawai[1,2,3], Ville Haapasilta[4], Benjamin D. Lindner[5], Kazukuni Tahara[3,5,6], Peter Spijker[4], Jeroen A. Buitendijk[7], Rémy Pawlak[2], Tobias Meier[2], Yoshito Tobe[5], Adam S. Foster[4,8] & Ernst Meyer[2]

On-surface chemical reactions hold the potential for manufacturing nanoscale structures directly onto surfaces by linking carbon atoms in a single-step reaction. To fabricate more complex and functionalized structures, the control of the on-surface chemical reactions must be developed significantly. Here, we present a thermally controlled sequential three-step chemical transformation of a hydrocarbon molecule on a Cu(111) surface. With a combination of high-resolution atomic force microscopy and first-principles computations, we investigate the transformation process in step-by-step detail from the initial structure to the final product via two intermediate states. The results demonstrate that surfaces can be used as catalysing templates to obtain compounds, which cannot easily be synthesized by solution chemistry.

[1] International Center for Materials Nanoarchitectonics, National Institute for Materials Science, 1-1, Namiki, Tsukuba, Ibaraki 305-0044, Japan. [2] Department of Physics, University of Basel, Klingelbergstrasse 82, Basel CH-4056, Switzerland. [3] Precursory Research for Embryonic Science and Technology (PRESTO), Japan Science and Technology Agency, 4-1-8,Honcho, Kawaguchi, Saitama 332-0012, Japan. [4] COMP, Department of Applied Physics, Aalto University, PO Box 11100, FI-00076 Aalto, Finland. [5] Department of Materials Engineering Science, Graduate School of Engineering Science, Osaka University, 1-3, Machikaneyama, Toyonaka, Osaka 560-8531, Japan. [6] Department of Applied Chemistry, School of Science and Technology, Meiji University, 1-1-1 Higashimita, Tama-ku, Kawasaki 214-8571, Japan. [7] Kantonsschule Trogen, Kantonsschulstrasse 20-29, CH-9043 Trogen, Switzerland. [8] Division of Electrical Engineering and Computer Science, Kanazawa University, Kanazawa 920-1192, Japan. Correspondence and requests for materials should be addressed to S.K. (email: KAWAI.Shigeki@nims.go.jp) or to V.H. (email: ville.haapasilta@aalto.fi) or to K.T. (email: tahara@meiji.ac.jp).

On-surface chemical reactions provide an attractive route for fabricating tailored nano-structures directly on surfaces. In the approach, the designed precursor molecules are deposited on surfaces in ultra-high vacuum conditions and the desired reactions are induced by thermal annealing. One of the main strengths in this approach is the possibility to produce novel compounds and materials, which cannot be synthesized or characterized by solution chemistry easily—or at all. Furthermore, the ultra-high vacuum conditions enable the usage of modern scanning probe techniques to follow the reaction stages with unprecedented detail.

Recent efforts in fabricating on-surface nano-structures have produced conjugated polymers[1,2], graphene nanoribbons[3,4] and two-dimensional sheets[5] via Ullmann[1–4] and Glaser couplings[6], or for example using dehydration and esterification of boronic acid[7]. These nanoscale products are ideal samples for conductance[2,8] and mechanical measurements[9,10] but can also serve as key elements in luminescence diodes[11]. However, the production yield is generally relatively low, presumably related to the complicated reaction mechanism. For instance, Cirera et al.[12] recently reported that the reaction temperature can tune the probability of the intermolecular and intramolecular reaction pathways. Nevertheless, the lack of control and an incomplete mechanistic understanding of the on-surface processes remain as the main challenges hindering further progress.

In addressing these challenges the state-of-the-art atomic force microscopy has an important role to play. With a CO functionalized tip[13] the on-surface molecular structures can be directly resolved via the onset of Pauli repulsion, routinely enabling an atomic-scale resolution. Consequently, the technique has been used successfully to characterize several reaction products in one-step on-surface chemical reactions[14] and intermediates in the dimerization[15]. Furthermore, a systematic observation of tetracyclic pyrazino [2,3–f] [4,7] phenanthroline, annealed on Au(111), reveals the regioselectivity for on-surface dehydrogenative aryl-aryl bond formation[16]. However, from the perspective of developments in solution chemistry, multi-step chemical reactions are required to manufacture complex, functional on-surface nano-structures.

Here, we present a thermally controlled sequential three-step chemical transformation of triangular dehydrobenzo annulene (tDBA) on a Cu(111) surface in ultra-high vacuum conditions. The chemical structures of the original, two intermediates and the final product are directly observed by high-resolution atomic force microscopy. The transformations are also followed by density functional theory computations, addressing the molecular details of the structures and revealing the energetics along the reaction path.

## Results

**Intact tDBA on Ag(111).** We chose the planar triangular dehydrobenzo[12]annulene molecule[17,18] (tDBA, $C_{24}H_{12}$) as a precursor species for the investigation. The tDBA molecule, which consists of three benzene rings joined by three acetylene moieties (Fig. 1a), is known to undergo various reactions, such as complexation with transition metals[19] and transannular cyclization by reduction[20]. Furthermore, it was recently reported that molecules with spatially integrated multiple acetylene moieties can produce condensed aromatic molecules[14] and π-conjugated oligomers[21] via transannular cyclization on metal surfaces. Thus, tDBA provides a promising starting point to investigate molecular transformations.

Figure 1b shows a high resolution atomic force microscopy (AFM) image of tDBA deposited on a clean Ag(111) surface at room temperature, imaged with a CO functionalized tip.

The inset shows the corresponding scanning tunnelling microscopy (STM) topography in which the benzene rings appear as equivalent maxima. On Ag(111) tDBA retains its intrinsic planar and triangular structure. Due to the tilt effect of the flexible CO tip[22] the benzene rings are observed as strongly distorted. The concentrated charge density in the acetylene moiety triple bonds is observed as bright spots (less negative frequency shift)[23]. On the whole, the tDBA molecule is sublimatable and stays intact on Ag(111) surface. This has been also demonstrated in another recent STM study on a Au(111) surface[24].

**First reaction on Cu(111) below 150 K.** Next, tDBA was deposited on a clean Cu(111) surface, which was kept at below 150 K. After the initial deposition the sample was annealed at 200 °C and 400 °C. At each stage the system was cooled and observed using AFM and STM at 4.8 K under ultra-high vacuum (UHV) conditions. Figure 2a shows an STM topography of tDBA as deposited on Cu(111) surface kept at below 150 K. In contrast to the deposition on Ag(111; Fig. 1b), no clear self-assembled pattern is observed on Cu(111). Furthermore, the molecule has adopted a new geometry and the original symmetric triangular structure has been lost. In addition to the isolated single molecules, the formation of dimers, trimers and larger structures is now observed. These more complex structures are more prominently observed when tDBA is deposited on a surface kept at room temperature (Supplementary Fig. 1). The shapes of the individual molecules are identical, whether or not they are observed as isolated single molecules or in larger agglomerations, or deposited on cold or warm surface. The inset in Fig. 2a shows a closer view of an isolated molecule: one bright (higher) and two dark (lower) spots can be clearly seen. Attempting to identify the structure, we imaged an individual molecule using an AFM with a CO functionalized tip (Fig. 2b). Similar to the STM topography, the AFM image shows one brighter feature (more positive frequency shift), suggesting that one part of the molecule is topographically higher than the others. Since the image was taken in constant height mode, only the protruded parts are directly observed[25]. Two, very faint, dark distorted circles seem to correspond to the position of the two benzene rings (indicated by blue arrows in Fig. 2b). To further investigate the transformed structure, we measured a three-dimensional frequency shift landscape[26] and consequently extracted the height at the most negative frequency shift point (Fig. 2c)[27]. The obtained image gives more structural information, since the tip can sense the lower part of the molecule—the height difference within the molecular structure was measured to be more than 200 pm. Assigning two benzene rings to the observed two local maxima at the bottom, we found that the most protruded parts are located on the initial acetylene moieties of the intact tDBA molecule, as indicated by the blue arrows in Fig. 2c. It seems that two acetylene bridges are partially reduced while two hydrogen atoms are added to each one[28]. The distance between the bottom benzene rings is ∼600 pm, which is close to the value of the intact tDBA (687 pm, Fig. 1a), suggesting that the third acetylene moiety is still intact. To gain more insight, we investigated the transformation using density functional theory (DFT) computations. The calculations reveal that there are several more stable configurations than the intact tDBA, where the two triple bonds have been transformed to double bonds by incorporating one hydrogen atom per bond. The most straightforward molecular structure after the transformation is shown in Fig. 2d (see also Supplementary Figs 2–5). The biradical molecule is stabilized by the interaction with the copper surface. In this proposed structure, which is 1.4 eV more stable than the intact tDBA on Cu(111), we suggest that two additional hydrogen atoms constitute the protruding

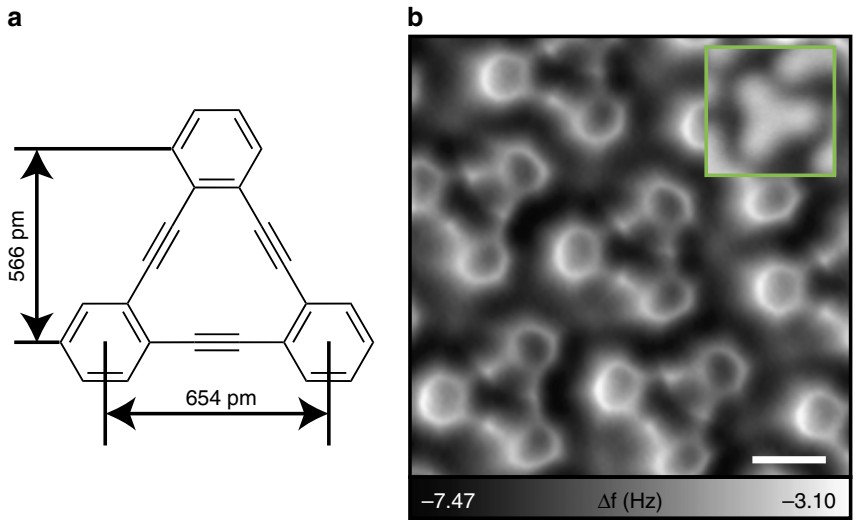

**Figure 1 | Intact tDBA on Ag(111).** (**a**) Chemical structure of triangular dehydrobenzo[12]annulene (tDBA). (**b**) High-resolution AFM image of tDBA on Ag(111). Inset shows the STM topography. Measurement parameters: $V_{tip} = 0$ mV and $A = 60$ pm in **b** and $V_{tip} = -200$ mV and $I = 1$ pA in (inset). Scale bar, 500 pm.

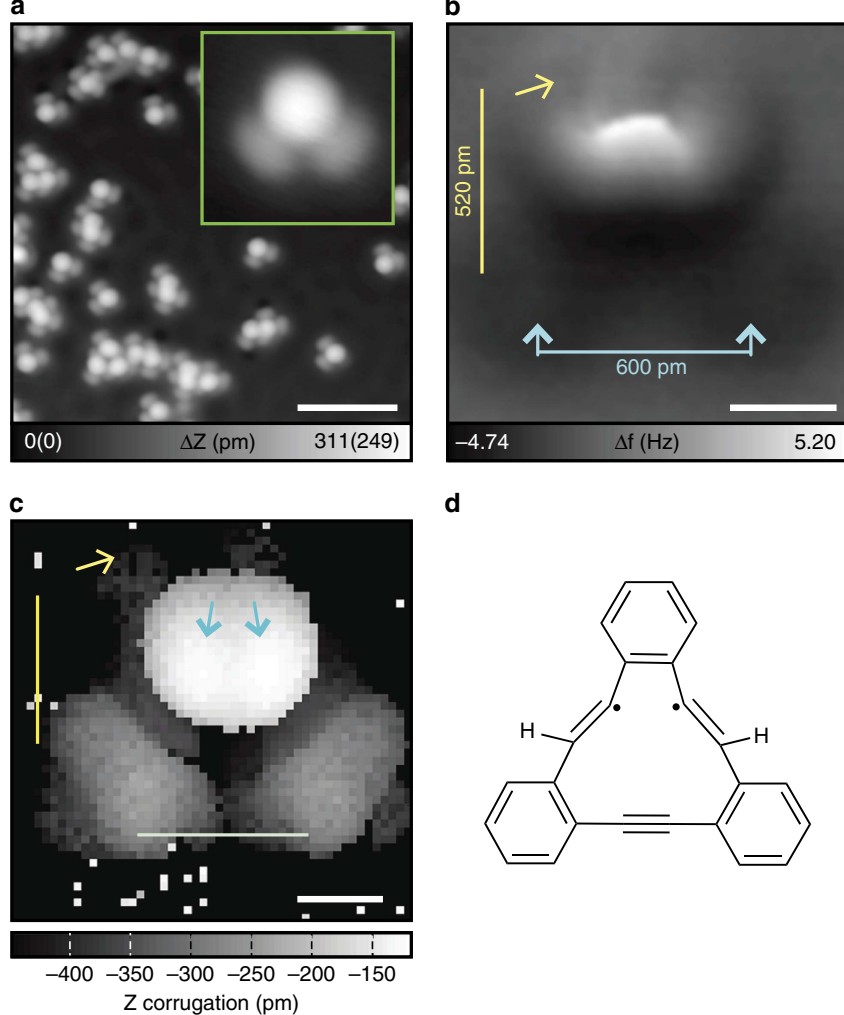

**Figure 2 | First reaction on Cu(111).** (**a**) Large-scale STM topography of tDBA as deposited on Cu(111) at 150 K. Inset shows the close view of the single molecule. (**b**) Corresponding AFM image, measured at a constant height. (**c**) Corrugation image, extracted via the three-dimensional frequency shift landscape. (**d**) Proposed chemical structure of the first product. Measurement parameters: $V_{tip} = -200$ mV and $I = 2$ pA in **a** and $V_{tip} = -200$ mV and $I = 1$ pA in (inset). $V_{tip} = 0$ mV and $A = 60$ pm in **b,c**. Scale bar, 4 nm in **a**. White, blue and yellow scale bars, 300 pm, 600 pm and 520 pm in **b,c**.

features observed in the STM and AFM images. Once the additional hydrogen atoms are incorporated into the molecular structure, they are not easily removed while the sample is held at lower than 150 K, which corresponds to a molecular kinetic energy of less than 1 eV. It should be noted that from an atomic point-of-view the reaction lowers entropy by confining diffusing hydrogen atoms from the surface into the molecular structure. Here this negative change in entropy is compensated by the strong enough electronic binding energy. In practice this renders the transformation irreversible—the reverse reaction barrier is significantly higher than the forward one. This is in accordance with the observed production yield of nearly quantitative conversion.

It is safe to conclude that the high reactivity of the Cu(111) surface induces this chemical reaction even at a low temperature of 150 K. This observation is in line with on-surface Ullmann type coupling reactions, where the reaction temperature on Cu(111) was also found to be lower than that on Ag(111) surface[29]. It should be noted that the reaction selectivity is very high (nearly 100%, no side-product could be observed in our experiment) as well, in contrast to previously reported on-surface transformations. This is attributed to a confined molecular backbone of tDBA which suppresses side reactions, and the few undesired products tend to fuse with each other, so our focus remains on the the monomer product.

**Second reaction by tandem cyclization at 200 °C.** The sample was taken out from the low temperature microscope and immediately annealed at 200 °C for 10 min before returning to the low temperature microscope. The STM topography of the cooled-down sample is shown in Fig. 3a, where single and clustered molecules are observed; the detailed molecular structure of the monomer can be seen in the inset. While one bright and two darker spots still remain, the apparent shape of the molecule has become flatter, indicating that the molecular structure was transformed in the annealing process. Since the molecule is now fairly flat again, all intra-molecular bonds are clearly resolved in the corresponding AFM image (Fig. 3b). By now, all the three acetylene moieties have been transformed and the molecular structure consists of two pentacyclic and four hexacyclic carbon rings. The rim of the molecule is strongly pronounced (less negative frequency shift) indicating that corners of three benzene rings are topologically higher than the center. Furthermore, in the AFM image 14 C-H bonds are clearly visible, implying that the

elemental composition is now $C_{24}H_{14}$—with two hydrogen atoms likely having been added during the process. This was also confirmed by DFT calculations: the molecular structure without additional hydrogen atoms in the pentagonal rings turns out to be drastically corrugated (Supplementary Fig. 6). Note that the second reaction product, benzo[a]indeno[2,1–c]fluorene, which belongs to the family of indenofluorenes[30], has not been synthesized so far.

According to DFT calculations, the second-step product is very stable on Cu(111), over 4 eV more stable than the first-step product, which in turn is 1.4 eV more stable than the intact tDBA. Up to this point, the sequence of transformations is energetically a downhill process with fairly moderate barriers. In fact, this product can also be seen if the sample was left at room temperature for 12 h.

Besides isolated single molecules, chain-like structures are also observed (Fig. 3a). These polymers are composed of the second-step products, which are linked to each other via C–Cu–C organometallic bonds formed by dehydrogenation (Supplementary Fig. 7). Note that the C–C links remain in the next reaction at higher temperature (Supplementary Figs 8 and 9). It is well-known that Cu adatoms diffuse on the Cu(111) surface at room temperature[31]. The chain-like structures are predominantly observed if the sample is left at room temperature for a longer period of time, as the probability for the molecule to interact with the diffusing Cu adatoms increases (Supplementary Fig. 10). However, isolated molecules are preferentially observed if the temperature of the substrate is kept at 200 °C during deposition (Supplementary Fig. 11). In this case, the thermal energy is large enough to form the second-step product with the help of Cu surface catalysis. These experimental results indicate that the reaction temperature has a significant impact on the formation process (Supplementary Fig. 12). The observed co-existing structures of the polymer and monomer in Fig. 3a result from these competing effects. Nevertheless, either monomer or polymer, the structure of all molecular units is the same as that described in Fig. 3c and so that no side-product was observed (Supplementary Figs 7 and 8).

**Third reaction by cyclodehydrogenation at 400 °C.** In the final step, the sample was further annealed at 400 °C for 30 min. The chain-like polymers aggregate on Cu(111), while some single molecules remain (Fig. 4a). The shape of the single molecule in the STM topography becomes rounder and flatter (see inset of

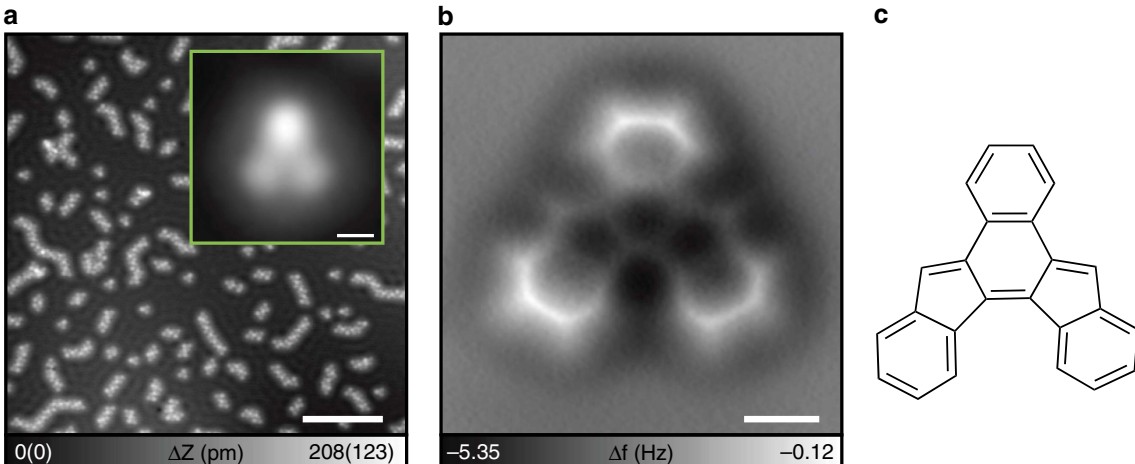

**Figure 3 | Second reaction by annealing at 200 °C.** (**a**) Large-scale STM topography. Inset shows the close view of the individual molecule. Scale bar (inset), 10 nm (500 pm). (**b**) AFM image of individual transformed tDBA. Scale bar, 300 pm. (**c**) Chemical structure of the product. Measurement parameters: $V_{tip} = -200$ mV and $I = 1$ pA in **a**. $V_{tip} = 0$ mV and $A = 60$ pm in **b**.

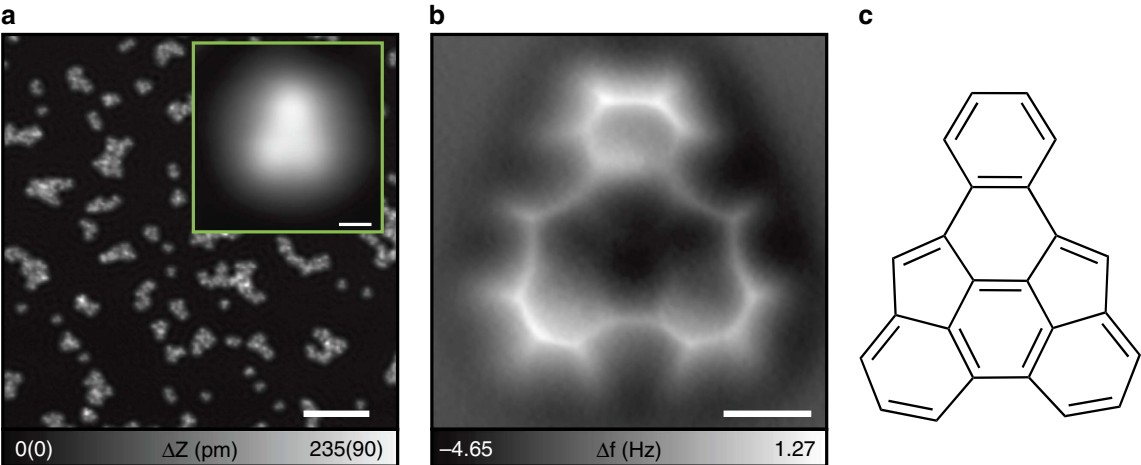

**Figure 4 | Third reaction by annealing at 400 °C. (a)** STM topography of tDAB on Cu(111) after annealing at 400 °C. Inset shows the close view. Scale bar (inset), 10 nm (500 pm). **(b)** AFM image, measured at constant height. Scale bar, 300 pm. **(c)** Chemical structure of the product. Measurement parameters: $V_{tip} = -200$ mV and $I = 10$ pA in **a**. $V_{tip} = 0$ mV and $A = 60$ pm in **b**.

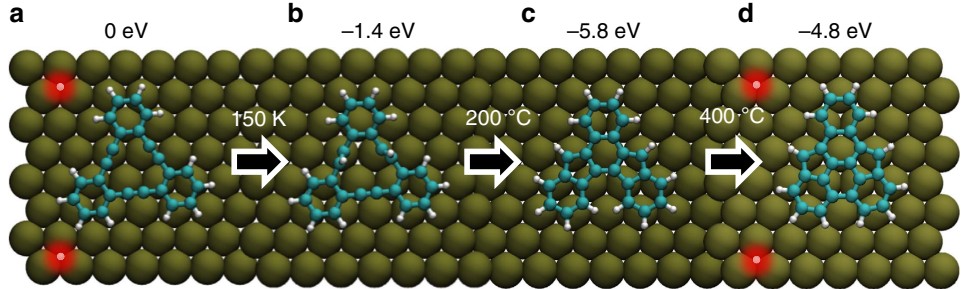

**Figure 5 | Summary of the transformation process.** The transformation proceeds from left to right (indicated by arrows). The reaction temperature for each transformation is shown. **(a)** Intact tDBA, which was not seen on Cu(111). **(b)** First product formed as deposited on a cold Cu(111). **(c)** Second product after annealing at 200 °C. **(d)** Third product after annealing at 400 °C. In the two middle structures the molecule contains 14 hydrogen atoms; in the first and last configurations these extra hydrogen atoms are placed on the copper surface (highlighted in red).

Fig. 4a), indicating that a further chemical reaction takes place. The corresponding AFM image shows the third-step product, where the molecular structure can be seen clearly (Fig. 4b). A new intramolecular bond between the two bottom benzene rings has been established. To form this C–C bond, two hydrogen atoms must be removed—thus the reaction proceeds via cyclodehydrogenation[32,33]. Since the newly formed C–C bond is much shorter (136 pm) than the gap between the bottom benzene rings in the second-step product (333 pm), the pentagonal and hexagonal carbon rings become slightly distorted, leading to a three-dimensional bowl-like structure. Energetically the third-step product is almost 5 eV more stable on Cu(111) than the intact tDBA molecule. It should be emphasized that external energy is required to have the last (stable) transformation: the transformation does not happen without annealing. This is reflected in the DFT calculations where the second-step product is about one electronvolt more stable than the third-step one. However, these are ground-state electronic energies of the optimized structures at 0 K. According to DFT calculations and first-principles molecular dynamics (FPMD) simulations, at 0 K hydrogen atoms adsorb as isolated atoms on the Cu(111) surface, but at 400 °C they form $H_2$, which subsequently desorbs from the surface. Thus, it is likely that in the last transformation the cleaved hydrogens form an $H_2$ molecule, which escapes into the vacuum. At high temperatures the role of entropy becomes more significant (Supplementary Table 1 and Supplementary Discussion). While sensitivity analysis from the FPMD

simulations shows that at 400 °C the difference in electronic energies in the last transformation is roughly 1.3 eV, the thermal contribution from the released $H_2$ molecule is more than 2.5 eV, rendering the final state more favourable in free energy (Supplementary Fig. 13). When the system is cooled down again for measurements, the molecule stays locked in the observed final configuration. Annealing at higher temperatures (>400 °C) breaks the molecular structures and induces an uncontrollable diffusion of the constituent carbon and hydrogen atoms[34]. This is in accordance with the melting temperature of coronene, roughlchy 440 °C, whi would be theoretically the most stable $C_{24}H_{12}$ end-product in the tDBA transformation process. We found that all isolated molecules transformed to the final product (Fig. 4c), which is again a new compound. Other polymeric molecules were formed by undefined in-plane carbon diffusion. These results indicate that the final product can be selectively collected from the surface by further thermal desorption if the molecules desorb before intermolecular reactions.

## Discussion
In summary, we report a thermally controlled three-step chemical reaction of triangular dehydrobenzo[12]annulene (tDBA) on a Cu(111) surface in ultra-high vacuum conditions (Fig. 5). We identify and investigate the transformation products of the annealing steps using a combination of high-resolution atomic

force microscopy and computations at the density functional level of quantum theory. We find that acetylene moieties can be induced to react by a suitable choice of surface and by heating. The unique confined backbone of the molecular structures— originating from the three-fold symmetric geometry of the intact tDBA—allows us to control the transformation process with annealing temperature, effectively leading to highly selective chemical reactions. The reactions produce two aromatic molecules, which have not been synthesized earlier (shown in Fig. 5c,d), to the best of our knowledge. We also note that the first two products could not be produced via conventional organic synthesis due to their reactivity and would decompose immediately in air. More specifically, the first product is an unstable radical species, and therefore the detection of such a species is a significant step for the chemical community. The second product benzo[a]indeno[2,1–c]fluorene, belongs to the family of indeno-fluorenes, an area of intense interest due to their unique properties[30].

In general, this demonstrates the potential of on-surface synthesis towards creating complex nano-structures: unlike in solution chemistry, in surface synthesis the desired molecules are formed directly on-surface, simplifying the process significantly. Together with previous studies, the results presented here make a strong case for high-resolution atomic force microscopy: it has become a very powerful tool to investigate on-surface chemical reactions, and especially when used in conjunction with first-principles computations, the technique is able to follow molecular transformations with unprecedented detail.

## Methods

**AFM measurements.** All measurements were performed with a commercially available Omicron low temperature STM/atomic force microscopy (AFM) system, operating in ultra-high vacuum at 4.8 K. We used a tuning fork with a chemically etched tungsten tip as a force sensor[35]. The resonance frequency and the mechanical quality factor are 24764.3 Hz and 23571, respectively. The high-stiffness of 1800 N m$^{-1}$ allows for stable operation with a small amplitude of 60 pm (ref. 36). The frequency shift, caused by the tip-sample interaction, was detected with a commercially available digital phase-locked loop (Nanonis: OC-4 and Zurich Instruments: HF2-LI and HF2-PLL)[37]. For the STM measurement, the bias voltage was applied to the tip while the sample is electronically grounded. The tip apex was *ex situ* sharpened by milling with a focused ion beam. The tip radius was <10 nm. A clean copper tip was *in situ* formed by indenting to the Cu sample surface and applying a pulse bias voltage between tip and sample several times. For AFM, the tip apex was terminated with a CO molecule, which was picked up from the surface[38]. Clean Ag(111) and Cu(111) surfaces were *in situ* prepared by repeated cycles of standard Ar$^{+}$ sputtering ($3 \times 10^{-6}$ mbar, 1,000 eV and 15 min) and annealing at 500 °C. tDBA was deposited on the surfaces from crucibles of a Knudsen cell, heated at 100 °C. The temperature of the substrate was kept below 150 K. After deposition, the sample was transferred to the microscope by a wobble stick manipulator. To minimize the influence of the heat transfer to the sample from the manipulator, the tweezers of the manipulator were cooled down by touching a helium radiation shield for ∼60 s and so that the temperature should be much lower than room temperature. Measured images were analysed using the WSxM software[39]. Three-dimensional frequency shift landscape was taken with drift-corrected dynamic force spectroscopy[26]. grid measurement points (51 × 51) were set over the molecule in the area of $1.4 \times 1.4$ nm$^2$. Individual Z distance-dependent measurements of frequency shift were performed at 256 equally spaced Z points over a 350-pm interval. The relative tip-sample position between each single Z distance-dependent measurements was readjusted by the atom tracking function.

**DFT calculations and first-principles molecular dynamics simulations.** All the (Kohn–Sham) density functional theory computations were performed using the Gaussian and plane wave method of the quickstep[40] module in the cp2k programme suite (www.cp2k.org). The PBE functional[41] with a D3 dispersion correction[42] was used in all of the calculations. The wavefunction was expanded using molecularly optimized Gaussian basis set (double zeta valence polarization (DZVP-molopt)[43]. The electronic density was described in terms of plane waves with a cutoff of 500Ry. The GTH pseudo-potentials[44] were used for the core electronic states, leaving eleven, four and one electron(s) to be considered explicitly for each Cu, C and H atom in the system, respectively. During each electronic step, the wavefunction was converged with an accuracy of $2.7 \times 10^{-5}$ eV or better.

Periodic boundary conditions were used. The size of the simulation box was $17.8476 \times 17.6645 \times 24.9814$ Å$^3$.

For the surface calculations, first a bulk Cu lattice constant had to be determined; the level of DFT used here gave a value of 3.60576 Å, which is within 0.25% of the experimental value[45]. The thickness of the suitable slab was determined by comparing the calculated surface energies to the experimental and theoretical literature values: a slab of six layers thick and $17.8476 \times 17.6645$ Å$^2$ wide gave a surface energy of 0.118 eV Å$^{-1}$, which is within $3.5 - 5.3$% of the experimental values[46–48]. The slab consisted of 336 Cu atoms. In the geometry optimization, calculations with molecules on the surface the convergence criteria was 0.01 eV Å$^{-1}$ or better. While optimizing the molecular structures in vacuum, the force gradient was converged better than 0.004 eV Å$^{-1}$. The basis set superposition error was estimated using the counterpoise method[49] with a wavefunction convergence better than $1 \times 10^{-5}$ eV. The adsorption energies given in the paper include the basis set superposition error correction.

In all of the Born–Oppenheimer FPMD simulations, a timestep of 0.5 fs was used. The simulations were performed in the canonical $NVT$ ensemble. The temperature was controlled with a Nosé-Hoover chain thermostat (coupling constant: 100 fs) attached to each cartesian degree of freedom[50].

**Data availability.** The data that support the findings of this study are available from the corresponding author on request.

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

## Acknowledgements

This work was supported in part by the Japan Science and Technology Agency (JST) 'Precursory Research for Embryonic Science and Technology (PRESTO)' for a project of 'Molecular technology and creation of new function', by the Swiss National Science Foundation, by the Swiss Nanoscience Institute, by COST Action MP1303 'Understanding and Controlling Nano and Mesoscale Friction', by the Academy of Finland through its Centres of Excellence Programme (2012–2017) under Project No. 251748, by EU project PAMS (contract no. 610446), by Postdoctoral Fellowship of Japan Society for the Promotion of Science (JSPS), by JSPS KAKENHI Grant Number 15K21765, and by the Swiss Academy of Sciences through the "Patenschaft für Maturaarbeiten" program. We gratefully acknowledge CSC—IT Center for Science Ltd., Espoo for computational resources.

## Author contributions

S.K. and J.A.B. conducted experiments. S.K. carried out data analysis. B.D.L., K.T. and Y.T. synthesized the precursor molecule. V.H. conducted theoretical computations. All contributed to writing the manuscript. S.K. and T.K. planned the project.

## Additional information

**Competing financial interests:** The authors declare no competing financial interests.

