## [Peer review file · Nature Communications]

Reviewers' Comments:

Reviewer #1 (Remarks to the Author)

A. Summary of key results.

The authors present a combined STM/nc-AFM/DFT study of the chemical transformations of a hydrocarbon precursor on Cu(111) upon thermal annealing identifying three sequential species, two of them unachievable by normal synthesis.

B. Originality and interest.

The paper is of interest for the surface science community and for the organic synthesis researchers. However, it is not so original for on-surface chemistry in view of two recent papers: i) Imaging single-molecule reaction intermediates stabilized by surface dissipation and entropy, *Nat Chem* 2016, and ii) Thermal selectivity of intermolecular versus intramolecular reactions on surfaces, *Nat. Commun.* 2016, which should be cited by the authors. Nevertheless, since there are almost no examples of a careful Nc-AFM identification of thermally evolved chemical compounds, I would recommend publication, though having in mind that this paper will not be the first of its kind.

C. Data & Methodology

The paper is brilliantly written and very easy to follow.

D. Appropriate use of statistics and treatment of uncertainties.

I am kind of worried because of two experimental limitations:

D.1. Instrumental. The molecules are deposited on a sample placed on a manipulator, held at a desired temperature. However, while transferring to a LT-STM omicron, there is a raise in temperature due to the fact that the wobble stick is at room temperature. Could the authors elaborate something on this respect?

D.2. The identification of the first chemical product after annealing should be relaxed in the manuscript. It is a tentative identification, not fully corroborated by nc-AFM. In addition, references addressing in-take of hydrogen atoms by molecules should be introduced.

D.3. Sublimation of last species from the surface could be something feasible or not...Too speculative.

E. Conclusions.

The paper is very solid and very appealing for the community.

F. Suggested improvements:

F.1. Distribution of distinct species upon steps of annealing. What are the side-products, if any?

F.2. Discussion on the mechanism of the reaction. Why these species are not possible by normal organic synthesis? What are the limiting steps? What impact do these species have for the organic community?

G.- References: The paper is missing the two Nature references mentioned above and the NanoLetters by A. Riss et al. (*Nano Lett.* 14, 2251-2255 (2014)).

H.- Clarity and context: Very clear and scholarly presented. Context should be slightly improved according to above mentioned suggestion.

Reviewer #2 (Remarks to the Author)

Interesting work which certainly can be recommended for publication in Nature Communication. The study addresses the topic of "on-surface" chemistry and it is timely to promote this issue for the physics community; it brings the two disciplines of chemistry and physics together to reach a common goal.

Some additional comments:

- 1) Abstract: The Abstract ends with "..., which cannot be synthesized by solution chemistry." This general statement is too strict. It should be changed to "... cannot easily be synthesized ...". In principle it is a matter of motivation and the resources one puts in from the chemistry side to synthesize a new compound.
- 2) Introduction: An additional Reference (J. Am. Chem. Soc, 2016, 138, 5585-5593 by Liu et al. Control of Reactivity and Regioselectivity for On-Surface Dehydrogenative Aryl-Aryl Bond Formation) needs to be cited. This recent paper goes very much in-line with the actual manuscript. A corresponding comment on it should be added in the Introduction.
- 3) Figure 2d: The two hydrogen atoms are bond with a line in a wedged (or tapered) form, which indicates a specific stereochemistry; it would be more appropriate to draw just a line.
- 4) First reaction: (page 6) "... these two triple bonds have been cleaved ..." I mean no, because then there would be no bonding anymore between the carbons. Better is for example, "Two acetylene bridges are partially reduced while two hydrogen atoms are added to each one."
- 5) (page 6) "The remaining two non-covalently bonded electrons are ...". This sounds strange for chemists; maybe: "The biradical molecule is stabilized by ...".
- 6) (page 8) Better would be: " By now, ... consists of two pentacyclic and four hexacyclic carbon rings."
- 7) (page 8) "... - two hydrogen atoms have definitely been added during the process". It sounds like: before maybe not but now for sure. Please rephrase this sentence.

Reviewer #3 (Remarks to the Author)

The paper reports on the thermally-controlled chemical transformation of tDBA on Cu(111) surface. The chemical structures of reaction intermediates and products were identified using the combination of STM, AFM, and DFT calculations. The study found two aromatic molecules unreported before, and the detailed mechanism of the three-step reaction was derived. In general, this paper is well-written with comprehensive data analysis. The data quality is excellent, but partial results are conceivable. I would recommend the paper for publication in Nature Communications after the authors address the following comments.

1. nc-AFM provides the essential information for determining the chemical structures of molecular species in the presented reaction. Although some interpretations appear to be straightforward, there are images hardly resembling the deduced structure (Fig.2b,c,d). Could the authors elaborate how the adsorption site of tDBT derivative was determined in DFT calculation (Fig.S2b)? In this spontaneous hydrogenation process, where do the hydrogen atoms come from? It is well known ethyne group readily react on copper surface by forming two C-Cu bonds, should this reaction path be considered as well?
2. In supplementary Fig.1, figure caption mentions (d) chemical structure of the trimer, which I cannot find. Additionally, it is mentioned "The observed bond-like feature in the intermolecular contact relates to an apparent bond, which is an artefact caused by the flexible CO tip following the potential energy landscape." I would like to ask for a clarification: whether the assembly of three DBA was bonded via C-H- π interactions OR they just got together by chance? If the argument of CO-tilting artefact prevails, I would doubt the validity of the approach of using AFM images to identify the intermediates or transition states in reactions. In view of the complicate configurations of the molecular moieties in the vicinity of catalyst atoms or active sites, should I believe those features in the acquired AFM images be real? Fig.S8 happens to be an example in this respect. The elongated bond length could arise from multiple origins; and the presence of a line feature is not a guarantee of the coordination bond that produces the organometallic complex. The bottom line for the above-mentioned questions: are the intra-molecular lines seen in Fig.3b,4b

resulted from tip artefact too?

3. Because of the absence of side reaction in the discussed system, the ratios of different products should agree well with the calculated thermodynamic energy. Could the authors have a comment on the issue?

4. Most organic molecules tend to break down at elevated temperatures on reactive metal surfaces like copper, why is it suggested that the final reaction product can be collected from the surface by further thermal desorption?

Reviewers' comments:

Reviewer #1 (Remarks to the Author):

A. Summary of key results.

The authors present a combined STM/nc-AFM/DFT study of the chemical transformations of a hydrocarbon precursor on Cu(111) upon thermal annealing identifying three sequential species, two of them unachievable by normal synthesis.

We thank Reviewer#1 for supporting our work and providing valuable comments to improve our manuscript. We carefully revised our manuscript accordingly.

B. Originality and interest.

The paper is of interest for the surface science community and for the organic synthesis researchers. However, it is not so original for on-surface chemistry in view of two recent papers: i) Imaging single-molecule reaction intermediates stabilized by surface dissipation and entropy, Nat Chem 2016, and ii) Thermal selectivity of intermolecular versus intramolecular reactions on surfaces, Nat. Commun. 2016, which should be cited by the authors. Nevertheless, since there are almost no examples of a careful Nc-AFM identification of thermally evolved chemical compounds, I would recommend publication, though having in mind that this paper will not be the first of its kind.

We agree with the comment that our work is not the first to report on on-surface chemical reactions. However, we believe that our work presents for the first time the sequential reaction of a single molecule on a surface. We now cite the two suggested articles. Please note that one of them (Nature Chem. 2016) appeared after our submission. We revised the manuscript on page 2 as

“However, the production yield is generally relatively low, presumably related to the complicated reaction mechanism. For instance, Cirera et al. recently reported that the reaction temperature can tune the probability of the intermolecular and intramolecular reaction pathways.¹²”

And on page 3

“and intermediates in the dimerization.¹⁵”

C. Data & Methodology

The paper is brilliantly written and very easy to follow.

We thank Reviewer#1 for supporting our work.

D. Appropriate use of statistics and treatment of uncertainties.

I am kind of worried because of two experimental limitations:

D.1. Instrumental. The molecules are deposited on a sample placed on a manipulator, held at a desired temperature. However, while transferring to a LT-STM omicron, there is a raise in temperature due to the fact that the wobble stick is at room temperature. Could the authors elaborate something on this respect?

We are familiar with this issue. To minimize the heat transfer, the tweezers of the wobble stick were cooled down by touching the cryostat for a while. For this reason, the temperature of the tweezers should be much lower than room temperature. Furthermore, the transfer to the microscope from the LT manipulator was usually done in less than 10 seconds (typically 3 seconds). To explain this, we added the following sentence in the experimental method section as

“After deposition, the sample was transferred to the microscope by a wobble stick manipulator. In order to minimize the influence of the heat transfer to the sample from the manipulator, the tweezers of the manipulator were cooled down by touching a helium radiation shield for about 60 seconds and so that the temperature should be much lower than room temperature.”

D.2. The identification of the first chemical product after annealing should be relaxed in the manuscript. It is a tentatively identification, not fully corroborated by nc-AFM. In addition, references addressing in-take of hydrogen atoms by molecules should be introduced.

We agree that the difficulty in fully imaging the first product makes our structural assignment somewhat tentative, despite the support of the simulations. We have modified the following sentence in the discussion (page 7) to make this clearer:

“In this proposed structure, which is 1.4~eV more stable than the intact tDBA on Cu(111), we suggest that two additional hydrogen atoms constitute the protruding features observed in the STM and AFM images.”

In a recent review article, Talirz *et al.* described the influence of the hydrogen atoms in the formation of the graphene nanoribbon. We have added the following reference to the manuscript:

L. Talirz, P. Ruffieux, and R. Fasel. On-surface synthesis of atomically precise graphene nanoribbons, *Adv. Mater.* (Early view) 2016. DOI: 10.1002/adma.201505738

D.3. Sublimation of last species from the surface could be something feasible or not...Too speculative.

As Reviewer#3 pointed out, we agree that it may be too speculative since we have never tried to collect the molecules. Yet, we are aware that the amount of the carbon decreased drastically during the polymerization by the dehydrogenation, meaning that the most of the molecule can be desorbed from the substrate. We revised the text on page 11 as

“These results indicate that the final product can be selectively collected from the surface by further thermal desorption if the molecules desorb before intermolecular reactions.”

E. Conclusions.

The paper is very solid and very appealing for the community.

We thank Reviewer#1 for supporting our work.

F. Suggested improvements:

F.1. Distribution of distinct species upon steps of annealing. What are the side-products, if any? The reaction yield to Product c (in figure 5) is 100%. At least in our measurements, we could not find any side-products. Yet, Product c tends to be connected with organometallic bonds as shown in Supplementary Figure 7 and 8. Once they are connected, the in-plane diffusion of the carbon atom in the polymeric molecules can happen by annealing at higher temperature (Supplementary Figure 9). Then, the side-products, namely polycyclic aromatic compounds, can be produced. In order to explain this, we added three sentences on page 7 as

“no side-product could be observed in our experiment”.

,on page 9 as

“Nevertheless, either monomer or polymer, the structure of all molecular units is the same as that described in Fig. 3c and so that no side-product was observed (Supplementary Fig. 7 and 8).”

,and on page 11 as

“Other polymeric molecules were formed by undefined in-plane carbon diffusion.”

F.2. Discussion on the mechanism of the reaction. Why these species are not possible by normal organic synthesis? What are the limiting steps? What impact do these species have for the organic community?

Why these species are not possible by normal organic synthesis? What are the limiting steps?

The first and second products are very reactive species, and immediately decomposed in air. Therefore, it is difficult or nearly impossible to produce these molecules by conventional organic synthetic methodologies. On the other hand, the third product, while it is a new molecule indeed, can be synthesized by the conventional procedures. To explain this, we added the following sentence on page 12:

“We also note that the first two products could not be produced via conventional organic synthesis due to their reactivity and would decompose immediately in air. “

What impact do these species have for the organic community?

As we mentioned above, the most important achievement in the present manuscript is the production and detection of unstable species by controlling *stepwise on-surface reactions*. The present information stimulates this research field further to produce novel molecules including very reactive species which have never been synthesized in solution chemistry. From an organic chemist's viewpoint, the first product is an unstable radical species, and therefore the detection of such a species has a significant impact on the community. Moreover, the second product, benzo[a]indeno[2,1-c]fluorene, which belongs to the family of indenofluorenes, is the subject of the interests for the organic chemists, because of its unique electronic and magnetic properties. As such, we believe our manuscript lead to strong impact on the organic community. To explain this, we added the following sentences on page 12

“More specifically, the first product is an unstable radical species, and therefore the detection of such a species is a significant step for the chemical community. The second product, benzo[a]indeno[2,1-c]fluorene, belongs to the family of indenofluorenes, an area of intense interest due to their unique properties.³⁰”

G.- References: The paper is missing the two Nature references mentioned above and the NanoLetters by A. Riss et al.(Nano Lett. 14, 2251-2255 (2014)).

The manuscript now cites these articles.

H.- Clarity and context: Very clear and scholarly presented. Context should be slightly improved according to above mentioned suggestion.

We thank Reviewer#1 for supporting our work.

Reviewer #2 (Remarks to the Author):

Interesting work which certainly can be recommended for publication in Nature Communication. The study addresses the topic of "on-surface" chemistry and it is timely to promote this issue for the physics community; it brings the two disciplines of chemistry and physics together to reach a common goal.

We thank Reviewer#2 for supporting our work. We carefully read the comments and revised our manuscript accordingly.

Some additional comments:

1) Abstract: The Abstract ends with "..., which cannot be synthesized by solution chemistry." This general statement is too strict. It should be changed to "... cannot easily be synthesized ...". In principle it is a matter of motivation and the resources one puts in from the chemistry side to synthesize a new compound.

We revised the last sentence as suggested.

2) Introduction: An additional Reference (J. Am. Chem. Soc, 2016, 138, 5585-5593 by Liu et al. Control of Reactivity and Regioselectivity for On-Surface Dehydrogenative Aryl-Aryl Bond Formation) needs to be cited. This recent paper goes very much in-line with the actual manuscript. A corresponding comment on it should be added in the Introduction.

We thank Reviewer#2 for suggesting this. We now cite this article in Introduction and added the following sentence on page 3 as

“Furthermore, a systematic observation of tetracyclic pyrazino [2,3-f][4,7]phenanthroline, annealed on Au(111), reveals the regioselectivity for on-surface dehydrogenative Aryl-Aryl bond formation.¹⁶”

3) Figure 2d: The two hydrogen atoms are bonded with a line in a wedged (or tapered) form, which indicates a specific stereochemistry; it would be more appropriate to draw just a line.

We modified Fig. 2d according to the reviewer's comment.

4) First reaction: (page 6) "... these two triple bonds have been cleaved ..." I mean no, because then there would be no bonding anymore between the carbons. Better is for example, "Two acetylene bridges are partially reduced while two hydrogen atoms are added to each one."

We revised as suggested.

5) (page 6) "The remaining two non-covalently bonded electrons are ...". This sounds strange for chemists; maybe: "The biradical molecule is stabilized by ...".

We revised as suggested.

6) (page 8) Better would be: " By now, ... consists of two pentacyclic and four hexacyclic carbon

rings."

We revised as suggested.

7) (page 8) "... - two hydrogen atoms have definitely been added during the process". It sounds like: before maybe not but now for sure. Please rephrase this sentence.

We have rephrased the sentence on page 8 as follows:

"Furthermore, in the AFM image 14 C-H bonds are clearly visible, implying that the elemental composition is now C₂₄H₁₄ -- with two hydrogen atoms likely having been added during the process."

Reviewer #3 (Remarks to the Author):

The paper reports on the thermally-controlled chemical transformation of tDBA on Cu(111) surface. The chemical structures of reaction intermediates and products were identified using the combination of STM, AFM, and DFT calculations. The study found two aromatic molecules unreported before, and the detailed mechanism of the three-step reaction was derived. In general, this paper is well-written with comprehensive data analysis. The data quality is excellent, but partial results are conceivable. I would recommend the paper for publication in Nature Communications after the authors address the following comments.

We thank Reviewer#3 for supporting our work and giving valuable comments on our manuscript. We carefully read and revised our manuscript accordingly.

1. nc-AFM provides the essential information for determining the chemical structures of molecular species in the presented reaction. Although some interpretations appear to be straightforward, there are images hardly resembling the deduced structure (Fig.2b,c,d). Could the authors elaborate how the adsorption site of tDBT derivative was determined in DFT calculation (Fig.S2b)? In this spontaneous hydrogenation process, where do the hydrogen atoms come from? It is well known ethyne group readily react on copper surface by forming two C-Cu bonds, should this reaction path be considered as well?

As Reviewer#3 points out, it is known that ethyne groups can react with copper to form C-Cu bonds and that was our initial guess as well. However, according to the performed DFT calculations (and first-principles molecular dynamics simulations) the formation of a new C-C bond within the molecule prevails, regardless of whether or not additional hydrogen atoms are introduced at this point. This is shown in Supplementary Information Fig. S4. If inspected carefully, from Fig.S4 one can see that there is indeed some interaction between the molecule and the copper surface, especially if no hydrogen atoms are added. However, this structure does not match with the corresponding AFM image. Indeed, only if one assumes that two hydrogen atoms are introduced to the molecule and it takes the form shown in Fig.S2a', does one obtain a qualitative agreement with the corresponding AFM image. We believe that the hydrogen atoms for the hydrogenation process come from the copper surface (please also see answers to Reviewer#1).

In the calculations a few initial adsorption sites on the surface were tried and the final geometries were obtained using a robust conjugate gradient optimization with a force gradient convergence better than 10 meV/Å.

2. In supplementary Fig.1, figure caption mentions (d) chemical structure of the trimer, which I cannot find. Additionally, it is mentioned "The observed bond-like feature in the intermolecular contact relates to an apparent bond, which is an artefact caused by the flexible CO tip following the potential energy landscape." I would like to ask for a clarification: whether the assembly of

three DBA was bonded via C-H- π interactions OR they just got together by chance? If the argument of CO-tilting artefact prevails, I would doubt the validity of the approach of using AFM images to identify the intermediates or transition states in reactions. In view of the complicated configurations of the molecular moieties in the vicinity of catalyst atoms or active sites, should I believe those features in the acquired AFM images be real? Fig.S8 happens to be an example in this respect. The elongated bond length could arise from multiple origins; and the presence of a line feature is not a guarantee of the coordination bond that produces the organometallic complex. The bottom line for the above-mentioned questions: are the intra-molecular lines seen in Fig.3b,4b resulted from tip artefact too?

With respect to the trimer, after discussion we agree that the evidence for its bonding is probably too speculative to be discussed in detail. Hence, we removed it from the SI. On the more general topic of bonding artefacts, this is where it is critical to support the AFM image with simulations - DFT agrees well with the Fig. 3b and Fig. 4b, and the intramolecular lines are not tip artifacts. This is already discussed in the SI in the case of Fig. 3.

3. Because of the absence of side reaction in the discussed system, the ratios of different products should agree well with the calculated thermodynamic energy. Could the authors have a comment on the issue?

We do not mean that there are no side reactions. The undesired products tend to fuse with each other. Thus, we count only the monomer product. To explain this, we revised the manuscript on page 7 as:

“It should be noted that the reaction selectivity is very high (nearly 100%, no side-product could be observed in our experiment) as well, in contrast to previously reported on-surface transformations. This is attributed to a confined molecular backbone of tDBA which suppresses side reactions, and the few undesired products tend to fuse with each other, so our focus remains on the the monomer product.”

4. Most organic molecules tend to break down at elevated temperatures on reactive metal surfaces like copper, why is it suggested that the final reaction product can be collected from the surface by further thermal desorption?

We agree with this comment. Please see the reply to Reviewer#1.

Reviewers' Comments:

Reviewer #1 (Remarks to the Author)

The points raised by referees in the previous round of review have been satisfactorily addressed and I am happy to recommend publication without further changes.

Reviewer #3 (Remarks to the Author)

The authors have revised the manuscript according to my comments. I feel that the paper is acceptable for Nature Communication.

Reviewers' comments:

Reviewer #1 (Remarks to the Author):

The points raised by referees in the previous round of review have been satisfactorily addressed and I am happy to recommend publication without further changes.

We thank Reviewer#1 for supporting our work.

Reviewer #3 (Remarks to the Author):

The authors have revised the manuscript according to my comments. I feel that the paper is acceptable for Nature Communication.

We thank Reviewer#3 for supporting our work.